Two new nematode species (Plectida: Leptolaimidae, Rhadinematidae) from Chatham Rise, New Zealand

Leduc Daniel daniel.leduc@gmail.com Daniel.Leduc@niwa.co.nz
National Institute of Water and Atmospheric Research (NIWA) , Wellington , New Zealand
Zhukova Natalia
Electronic publication date: 2020 Sep 11
Publication date: 2020
Volume: 8
Electronic Location ID: e9923
Received 2020 Jul 17; Accepted 2020 Aug 21
Copyright: ©2020 Leduc
Copyright year: 2020
Copyright holder: Leduc
License: This is an open access article distributed under the terms of the Creative Commons Attribution License, which permits unrestricted use, distribution, reproduction and adaptation in any medium and for any purpose provided that it is properly attributed. For attribution, the original author(s), title, publication source (PeerJ) and either DOI or URL of the article must be cited.
License URL: https://creativecommons.org/licenses/by/4.0/

Keywords: Continental slope, Chatham Rise, Free-living nematodes, Taxonomy

Funding: NIWA’s Coasts and Oceans Centre Research Programme ‘Marine Biological Resources’ FRST UOOX0909 “Consequences of Earth-Ocean Change” C01X0702 “Coasts & Oceans OBI” C01X0501 Research, Science and Technology (FRST) programme ‘Ocean Ecosystems’ C01X0027 New Zealand Ocean Survey 20/20 Chatham-Challenger projects funded by Land Information New Zealand, Ministry of Fisheries, Department of Conservation and NIWA Funding was provided by NIWA’s Coasts and Oceans Centre Research Programme ‘Marine Biological Resources’. Funding was also provided by FRST through a postdoctoral fellowship (UOOX0909), the programmes “Consequences of Earth-Ocean Change” (C01X0702), and “Coasts & Oceans OBI” (C01X0501). Sample data were generated under the Foundation for Research, Science and Technology (FRST) programme ‘Ocean Ecosystems’ (C01X0027) and the New Zealand Ocean Survey 20/20 Chatham-Challenger projects funded by Land Information New Zealand, Ministry of Fisheries, Department of Conservation and NIWA. The funders had no role in study design, data collection and analysis, decision to publish, or preparation of the manuscript.

==============================
Two new species of the order Plectida are described from Chatham Rise, New Zealand. Leptolaimus dififtinus sp. nov. is characterised by the short body 319–420 microns long, truncate labial region slightly offset from body contour and bearing conspicuous outer labial papillae, cephalic setae 1.3–1.4 microns long, amphid located 4–9 microns from anterior end, lateral alae originating from middle of buccal cavity length, female without supplements, male with precloacal and postcloacal pairs of subventral setae, nine tubular supplements (alveolar supplements absent), tubular supplements weakly S-shaped with pointed tip, spicules arcuate 24 microns or 1.4 cloacal body diameters long and dorsal gubernacular apophyses. Lavareda iramscotti sp. nov. is characterised by adult body length 3,023–3,121 microns long, eight longitudinal rows of body pores each with short papilla, cephalic setae 4–5 microns long, tail 146–165 microns long, male with spicules 54 microns long, gubernaculum with triangular apophyses, 20 precloacal supplements with bifid distal tips arranged in 9 + 1 + 10 pattern, female with vulva at 55% of body length from anterior and cuticularisation perpendicular to vagina at level of vulva. The present study provides the first record of a Leptolaimus species from the New Zealand region and the first description of a female specimen of the genus Lavareda.

Introduction

The order Plectida Gadea, 1973 comprises nematode taxa inhabiting marine, brackish, freshwater and soil habitats, as well as a few commensal and potentially parasitic taxa (Holovachov, 2014). To date, only six marine plectid species have been recorded from New Zealand waters by Allgén (1927), Allgén (1932), Ditlevsen (1930) and Leduc & Wilson (2016). The family Leptolaimidae Örley, 1880 is a globally distributed, largely marine and brackish family with Leptolaimus De Man, 1876 one of most common and diverse plectid genus. Sixty valid Leptolaimus species are currently known (Holovachov & Boström, 2013; Tchesunov, 2015; Qiao, Jia & Huang, 2020), but no Leptolaimus species have yet been recorded from the New Zealand region.

The Rhadinematidae Lorenzen, 1981, is a small, rare group of marine plectids comprising three genera and five valid species (Holovachov, 2014; Holovachov & Boström, 2014). Two genera, viz., Cricolaimus Southern, 1914 and Lavareda Da Fonsêca-Genevois, Smol & Bezerra, 2011 are so far known only from males. One species, Lavareda coronatus (Ditlevsen, 1930) Da Fonsêca-Genevois, Smol & Bezerra, 2011 has been recorded from the New Zealand region. This species originally belonged to the genus Cricolaimus but was transferred to Lavareda based on the structure of the cheilostom (Da Fonsêca-Genevois, Smol & Bezerra, 2011). Here, two new species of the order Plectida are described from Chatham Rise, New Zealand: Leptolaimus dififtinus sp. nov. and Lavareda iramscotti sp. nov.

Materials & Methods

Samples were obtained from Chatham Rise, a submarine ridge that extends eastwards from the South Island of New Zealand, over water depths ranging from ca. 250 to 3,000 m. Samples were collected from Chatham Rise during NIWA cruise TAN0705 (March–April 2007) and TAN1103 (February 2011) under Special Permit 666 granted to NIWA by the Ministry for Primary Industries. Sediment samples were collected and processed as described in Leduc (2012), using an Ocean Instruments MC-800A multicorer (core internal diameter = 9.5 cm). Each sample consisted of one subcore of internal diameter 26 mm taken to a depth of 5 cm. Samples were fixed in 10% formalin and stained with Rose Bengal. Samples were subsequently rinsed on a 45 µm sieve to retain nematodes. Nematodes were extracted from the remaining sediments by Ludox flotation and transferred to pure glycerol (Somerfield & Warwick, 1996).

Species descriptions were made as described in Leduc (2020), based on glycerol mounts using differential interference contrast microscopy and drawings were made with the aid of a camera lucida. Measurements were obtained using an Olympus BX53 compound microscope with cellSens Standard software. All measurements are in µm, and all curved structures are measured along the arc. The terminology used for describing the arrangement of morphological features such as setae follows Coomans (1979). Type specimens are held in the NIWA Invertebrate Collection (Wellington).

The electronic version of this article in Portable Document Format (PDF) will represent a published work according to the International Commission on Zoological Nomenclature (ICZN), and hence the new names contained in the electronic version are effectively published under that Code from the electronic edition alone. This published work and the nomenclatural acts it contains have been registered in ZooBank, the online registration system for the ICZN. The ZooBank LSIDs (Life Science Identifiers) can be resolved and the associated information viewed through any standard web browser by appending the LSID to the prefix http://zoobank.org/. The LSID for this publication is: urn:lsid:zoobank.org:pub:4689626F-D42C-464A-BB0B-55D593DD3E71. The online version of this work is archived and available from the following digital repositories: PeerJ, PubMed Central and CLOCKSS.

Results

Order Plectida Gadea, 1973	
Family Leptolaimidae Örley, 1880	
Genus LeptolaimusDe Man, 1876	
=HalaphanolaimusSouthern, 1914	
=AplectusCobb, 1914	
=DermatolaimusSteiner, 1916	
=PolyaimiumCobb, 1920	
=AlveolaimusAlekseev & Rassadnikova, 1977	
=BoveelaimusAlekseev, 1979	
=TubulaimusAlekseev & Rassadnikova, 1977	

Generic diagnosis: (from Holovachov (2014)) Lateral alae present. First annule anterior to cephalic setae bases and amphids, cephalic capsule absent. Cephalic sensilla papilliform or setiform. Amphideal aperture ventrally unispiral, without central elevation. Secretory-excretory system present; excretory canal short, excretory ampulla present. Ovary branches reflexed antidromously. Alveolar or tubular supplements present in females of some species, either in pharyngeal or pre-anal regions, or in both positions. Male reproductive system diorchic. Number of supplements varies from zero to 40 for alveolar and zero to 11 for tubular; males may have both types of supplements, one or no supplements at all. Caudal glands and spinneret present or absent.

Type species: L. papilliger De Man, 1876

Remarks. The genus was revised by Holovachov & Boström (2013) who provided a key to the identification of all 58 valid species based on features of both males and females. Two species were subsequently described by Tchesunov (2015) and Qiao, Jia & Huang (2020).

Leptolaimus dififtinus sp. nov.	
Figs. 1, 2, 3 and 4, Table 1	
urn:lsid:zoobank.org:act:C4FE8E90-6481-4E1E-9780-561C4D1D8D9E	

Type locality: Northern flank of Chatham Rise, 1194 m water depth (RV Tangaroa voyage TAN0705, station 196, site D15; 42.6147°S, 178.3338°W). Muddy sand sediments (68% sand, 19% silt, 13% clay), calcium carbonate content 59.9%.

Type material: Holotype male (NIWA 139251), and one paratype female (NIWA 139252), collected on 19 April 2007.

Measurements: See Table 1 for detailed measurements.

Description: Male. Body colourless except for dense brown granules in anterior portion of intestine, fusiform, strongly curved ventrally (maybe as a result of formalin fixation), strongly tapering anteriorly in pharyngeal region and posteriorly on tail. Cuticle annulated, annules ca. 1.3 µm apart; lateral alae present consisting of raised, non-annulated cuticle ca. 2.5–3.0 µm wide extending from middle of buccal cavity length (close to first body pore) to almost half of tail length. Four longitudinal rows of sublateral body pores, extending from middle of buccal cavity to two thirds of tail length, pores in pharyngeal region with short papilla, pores posterior to pharynx without papillae; epidermal glands not observed. Somatic setae absent except for a pair of subventral setae, 2 µm long, anterior to cloaca, and second pair of subventral setae posterior to cloaca. Labial region truncate, slightly offset from body contour, lips fused. Inner labial sensilla indistinct; outer labial sensilla conspicuous and papillose, located on the outer surface of lips. Cephalic sensilla setiform, 0.25–0.30 cbd long. Ocelli absent. Amphideal fovea round, located at level of anterior quarter of buccal cavity. Buccal cavity uniformly tubular: cheilostom and gymnostom short, undifferentiated; stegostom tubular, with uniformly thickened lumen. Pharynx muscular, cylindrical anteriorly, with distinct oval basal bulb, distinctly cuticularized lumen from posterior to buccal cavity to near posterior end of pharynx; valvular apparatus absent. Pharyngeal glands and their orifices indistinct. Nerve ring surrounding pharynx near middle of pharynx length. Secretory-excretory system not observed. Cardia cylindrical, with posterior part embedded in intestine.

Reproductive system diorchic with both testes directed anteriorly; one larger outstretched testis located to the right of intestine with globular sperm ca. 2 µm in diameter, one smaller outstretched (possibly reflexed) testis located ventrally with smaller globular sperm ca. 1 µm in diameter. Spicules paired, symmetrical, arcuate, 1.4 cloacal body diameter long, only slightly cuticularised; capitulum rounded, shaft and blade gradually tapering distally. Gubernaculum with lightly cuticularized, bent dorsal apophysis. Ventral precloacal seta not observed. Accessory apparatus composed of nine more or less evenly spaced midventral tubular supplements (posterior supplements appear closer together due to body curvature), ca. 16–23 µm apart, extending for 8 µm from cloaca towards anterior end; alveolar supplements absent. Tubular supplements weakly S-shaped, with pointed tip, 11–14 µm long, each with small drop-shaped subsurface cuticularisation immediately anterior. Tail conical; three caudal glands and spinneret present.

Figure 1 Leptolaimus dififtinus sp. nov.

(A & C) Anterior body region of male; (B) anterior body region of female; (D) posterior body region of female; (E) posterior body region of male. Scale bar: A, B, C & D = 20 μm, E = 25 μm.

Figure 2 Leptolaimus dififtinus sp. nov.

(A) Entire male; (B) entire female. Scale bar: A = 25 μm, B = 20 μm.

Figure 3 Leptolaimus dififtinus sp. nov. light micrographs.

(A) Optical cross section of entire female; (B) surface view of female; (C) optical cross-section of entire male; (D) surface view of entire male. Scale bar = 20 microns.

Figure 4 Leptolaimus dififtinus sp. nov. light micrographs of male.

(A) Optical cross-section of pharyngeal region; (B) surface view of cephalic region; (C) optical cross-section of precloacal supplements; (D) tail. Scale bar: A & B = 10 microns, C & D = 11 microns.

Female. Similar to male, but without any somatic setae, lower values of ‘a’ and shorter tail. Reproductive system with single anterior reflexed ovary located ventrally; posterior ovary likely present but obscured by large egg in uterus. Spermathecae not observed. Vulva situated slightly posterior to mid-body. Vagina slightly anteriorly directed, short, proximal portion encircled by single sphincter muscle. Vaginal glands not observed. Supplements absent.

Diagnosis: Leptolaimus dififtinus sp. nov. is characterised by short body 319–420 µm long, truncate labial region slightly offset from body contour and bearing conspicuous outer labial papillae, cephalic setae 1.3–1.4 µm long, amphid located 4–9 µm from anterior end, lateral alae originating from middle of buccal cavity length, female without supplements, male with precloacal and postcloacal pairs of subventral setae, nine tubular supplements (alveolar supplements absent), tubular supplements weakly S-shaped with pointed tip, spicules arcuate 24 µm or 1.4 cloacal body diameters long and dorsal gubernacular apophyses.

Table 1 Morphometrics (µm) of Leptolaimus dififtinus sp. nov. and Lavareda iramscotti sp. nov.

	Leptolaimus dififtinus sp. nov.	Lavareda iramscotti sp. nov.	
	Male	Female	Male	Female	Juvenile 1	Juvenile 2	
	Holotype	Paratype	Holotype	Paratype	Paratype	Paratype	
L	430	319	3,023	3,121	2,491	2,846	
a	20	12	104	82	92	66	
b	4	4	17	17	16	15	
c	8	8	18	21	19	19	
c′	3.4	2.9	5.3	5.2	5.1	5.0	
Head diam. at cephalic setae	5	5	10	11	11	11	
Head diam. at amphids	8	7	18	20	18	20	
Length of cephalic setae	1.3–1.4	1.3	4.5	4.4	5.5	4.5	
Amphid height	4	3	8	7	8	9	
Amphid width	4	4	9	8	8	7	
Amphid width/cbd (%)	50	57	50	40	44	45	
Amphid from anterior end	9	4	21	19	18	21	
First body pore from anterior	24	22	40	36	36	39	
Lateral alae from anterior	24	33	–	–	–	–	
Buccal cavity length	41	34	28	30	26	29	
Nerve ring from anterior end	67	52	94	84	82	82	
Nerve ring cbd	20	18	26	29	29	34	
Pharynx length	110	90	176	181	160	185	
Pharyngeal bulb length	19	17	28	31	25	32	
Pharyngeal bulb diam.	13	16	17	20	19	23	
Pharynx cbd	22	20	29	34	32	38	
Max. body diam.	22	27	29	38	27	43	
Spicule length	24	–	54	–	–	–	
Gubernaculum length	4	–	11	–	–	–	
Cloacal/anal body diam.	17	13	31	28	25	30	
Tail length	57	38	165	146	128	150	
V	–	174	–	1,725	–	–	
%V	–	55	–	55	–	–	
Vulval body diam.	–	26	–	35	–	–	
Notes.

a body length/maximum body diameter

b body length/pharynx length

c body length/tail length

c′ tail length/anal or cloacal body diameter

cbd corresponding body diameter

L total body length

n number of specimens

V vulva distance from anterior end of body

%V V/total body length × 100

Differential diagnosis: The new species is similar to Leptolaimus alatus Vitiello, 1971, L. sextus Holovachov & Boström, 2013 and L. septimus Holovachov & Boström, 2013 particularly in having distinct outer labial sensilla and a labial region offset from body contour, but also in males having relatively short spicules (less than three cloacal body diameters), four or more weakly S-shaped tubular supplements without distal discs and in a continuous row, alveolar supplements absent, and females without supplements. Leptolaimus dififtinus sp. nov. is most similar to L. alatus but can be differentiated from the latter by the slightly shorter cephalic setae (1.3–1.4 µm or 0.25–0.30 cbd versus 1.5–2.0 µm or 0.30–0.50 cbd), anterior body pores with short papillae (versus long setae in L. alatus), number of precloacal supplements (9 versus 6–8 supplements in L. alatus) and shape of the gubernacular apophysis (bent and dorsally-directed versus plate-like and dorsocaudally-directed in L. alatus). In the original description of L. alatus by Vitiello (1971) based on specimens from the Mediterranean, the supplements have pointed tips (as in L. dififtinus sp. nov.), however Holovachov & Boström (2013) describe the tips as blunt in their L. alatus specimens from Skagerrak and Swedish fjords.

Leptolaimus dififtinus sp. nov. can be differentiated from L. sextus by the shorter body length (319–420 versus 626–728 µm in L. sextus), anterior body pores with short papillae (versus long setae in L. sextus), shorter spicules (24 versus 39–46 µm in L. sextus), and number and shape of precloacal supplements (9 supplements with pointed tips versus 5–6 supplements with bifid tips in L. sextus). Leptolaimus dififtinus sp. nov. can be differentiated from L. septimus by the shorter body length (319–420 versus 679–821 µm in L. septimus), anterior body pores with short papillae (versus long setae in L. septimus), lower ‘a’ ratio (12–20 versus 24–37 in L. septimus), shorter cephalic setae (1.3–1.4 µm versus 2.5–3.5 µm in L. septimus), smaller amphid (4 versus 5-6 µm wide in L. septimus), shorter tail (2.9–3.4 versus 4.7–7.1 cloacal/anal body diameters long in L. septimus), shorter spicules (24 versus 31-34 µm in L. septimus) and number and shape of precloacal supplements (9 supplements with pointed tips versus 4-5 supplements with bifid tips in L. septimus).

Etymology: The species name is an arbitrary combination of letters (ICNZ Article 11.3) and refers to the sampling site (D15) where the type specimens were collected.

Family Rhadinematidae Lorenzen, 1981	
Genus LavaredaDa Fonsêca-Genevois, Smol & Bezerra, 2011	

Generic diagnosis: (modified from Da Fonsêca-Genevois, Smol & Bezerra (2011) and Holovachov (2014)) Body long, cylindrical, slender. Cuticle annulated, lateral alae absent. Epidermal glands and body pores present or absent. Lips low, inner and outer labial papilla indistinct. Cheilostom consists of a cuticularised ring with six anteriorly directed projections (one dorsal, one ventral, and four sublateral in position). Amphideal aperture loop-shaped, ventrally unispiral. Buccal cavity funnel-shaped to tubiform. Pharynx cylindrical with posterior oval bulb, muscular. Cardia embedded in intestine. Male reproductive system diorchic or monorchic, testes directed anteriorly. Spicules relatively short, strongly curved, manubrium enlarged, gubernaculum with apophysis. Nineteen or twenty midventral precloacal tubular supplements, precloacal seta present or absent. Female without supplements.

Type species: L. decraemerae Da Fonsêca-Genevois, Smol & Bezerra, 2011

Lavareda iramscotti sp. nov.	
Figs. 5, 6 and 7, Table 1	
urn:lsid:zoobank.org:act:79592D96-3AAE-4B67-9D23-14A158799216	

Type locality: Chatham Rise crest, 347 m water depth (RV Tangaroa voyage TAN1103, station 69, 41.3352°S, 178.2962°E). Sandy mud sediments (44% sand, 55% silt, 1% clay).

Type material: Holotype male (NIWA 139253), one paratype female and two paratype juveniles (NIWA 139254), collected on 2 February 2011.

Measurements: See Table 1 for detailed measurements.

Description: Male. Body colourless in glycerin preparations, long, slender, uniform in diameter throughout most of body, tapering towards both ends. Cuticle annulated from posterior to cephalic setae to spinneret, annules ca. 1.2 µm apart, without ornamentation or lateral differentiation. Eight longitudinal rows of body pores (two subdorsal, two subventral and four sublateral) extending from posterior to buccal cavity to tail region; each pore with a short conical papilla and connected to small, barely visible epidermal gland. Cephalic region slightly rounded, with reduced labial region. Inner labial sensilla indistinct, outer labial sensilla papilliform, four cephalic setae 0.4–0.5 cbd long. Amphideal fovea ca. 1 cbd from anterior extremity, ventrally unispiral; corpus gelatum not observed. Buccal cavity narrow, deep, tubular, without teeth; cheilostom distinct, resembling a crown, 5 µm in diameter, consisting of a cuticularized ring with six small anteriorly-directed projections (one dorsal, one ventral and four lateral in position). Pharynx muscular, gradually widening posteriorly from posterior to buccal cavity and ending in an oval posterior bulb; pharyngeal lumen cuticularized from posterior to buccal cavity and most thickly cuticularized within posterior pharyngeal bulb. Nerve ring located slightly posterior to middle of pharynx length. Cardia small, 8 µm long, embedded within intestine. Secretory-excretory system with slightly cuticularized pore located posterior to nerve ring and small ampulla; four pairs of medium to large glands with clear cytoplasm located on each side of intestine, up to 33 × 16 µm, starting from ca. 1–2 cbd posterior to pharynx, followed by two pairs of glands with more opaque cytoplasm immediately posteriorly, each gland with duct extending anteriorly.

Reproductive system with two anteriorly directed testes, anterior testis outsretched, 430 µm long, located to the right of intestine, posterior testis reflexed, 180 µm long, located ventrally relative to intestine. Mature sperm cells globular, dimorphic; 14–20 × 11–15 µm in anterior testis, 6–7 × 4–5 µm in posterior testis. Spicules paired, equal, 1.7 cloacal body diameters long, strongly ventrally curved and narrowing distally, with well-developed capitulum; gubernaculum strongly cuticularized with triangular, dorsocaudally directed apophyses. Twenty tubular precloacal supplements, ca. 22 µm long, swollen proximally and with bifid tip, each connected to a gland near proximal end and extending anteriorly; posterior group of nine supplements situated gradually further apart anteriorly (19–52 µm distance), posterior group of ten supplements more or less equidistant from each other (39–53 µm distance), and single supplement located between the posterior and anterior groups of supplements. Precloacal seta present. Tail conical, curved ventrally, with three caudal glands and well-developed spinneret.

Female. Similar to male, but with lower ‘a’ ratio and smaller amphid. Reproductive system with two opposed, reflexed ovaries; anterior ovary situated to the right of intestine and posterior ovary situated to the left of intestine. Spermathecae not observed. Vulva situated slightly posterior to mid-body. Vagina straight, with thick cuticle, two vaginal glands present. Thin cuticularisation centred at level of vulva and perpendicular to vagina, ca. 1 vulval body diameter long. Sphincter muscle not observed. Supplements absent.

Juveniles. Similar to female but body slightly shorter and ‘ c′’ ratio slightly lower. Genital primordium present.

Diagnosis: Lavareda iramscotti sp. nov. is characterised by adult body length 3023-3121 µm, eight longitudinal rows of body pores each with a short papilla, cephalic setae 4–5 µm or 0.4–0.5 cbd long, tail 146–165 µm or 5.2–5.3 cloacal/anal body diameters long, male with spicules 54 µm or 1.7 cbd long, gubernaculum with triangular apophyses, 20 precloacal supplements with bifid distal tips arranged in 9 + 1 + 10 pattern, female with vulva at 55% of body length from anterior and cuticularisation perpendicular to vagina at level of vulva.

Figure 5 Lavareda iramscotti sp. nov.

(A) Anterior body region of male; (B) female cephalic region; (C) juvenile cephalic region; (D) male cephalic region; (E) male posterior body region; (F) female anterior body region; (G) female posterior body region. Scale bar: A = 50 μm, B, C & D = 20 μm, E = 45 μm, F = 60 μm, G = 40 μm.

Figure 6 Lavareda iramscotti sp. nov.

(A) Entire male; (B) female reproductive system. Scale bar: A = 75 μm, B = 185 μm.

Figure 7 Lavareda iramscotti sp. nov. light micrographs of male.

(A) Lateral surface view of anterior body region; (B) optical cross-section of anterior body region; (C) precloacal supplement; (D) copulatory apparatus; (E) tail tip and spinneret. Scale bar: A & B = 10 μm, C & E = 12 μm, D = 14 μm.

Differential diagnosis: The new species differs from the two other Lavareda species, L. coronatus (Ditlevsen, 1930) Da Fonsêca-Genevois, Smol & Bezerra, 2011 and L. decraemerae, by the presence of longitudinal rows of body pores (versus pores absent in L. coronatus and L. decreamerae). Additional differences between the male of L. iramscotti sp. nov. and L. decraemerae are: greater body length (3.0 versus 2.5 mm in L. decraemerae), shorter cephalic setae (4–5 versus 9 µm in L. decraemerae), absence of cervical setae (versus eight cervical setae in L. decraemerae), longer spicules (54 versus 30 µm in L. decraemerae), and number and arrangement of precloacal supplements (20 supplements in 9 + 1 + 10 arrangement versus 19 supplements in continuous row in L. decraemerae). Furthermore, the secretory-excretory system in L. decraemerae comprises five glands, two of which are located on the right of the intestine and three on the left side of the intestine; Lavareda iramscotti sp. nov. is characterised by an secretory-excretory system with ten glands, five located on either side of the intestine. Lavareda iramscotti sp. nov. can be differentiated from L. coronatus by the higher ‘a’ ratio (104 versus 63 in L. coronatus), higher ‘b’ ratio (17 versus 14 in L. coronatus), lower ‘c’ ratio (18 versus 24 in L. coronatus), presence of cephalic setae (versus absent in L. coronatus), longer tail (165 versus 114 µm in L. coronatus), male reproductive system with two testes (versus one testis in L. coronatus), presence of precloacal seta (versus absent in L. coronatus), and arrangement of precloacal supplements (9 + 1 + 10 versus 9 + 11 arrangement in L. coronatus).

Etymology: The species name is an arbitrary combination of letters (ICNZ Article 11.3) which acknowledges my colleague Scott Nodder, who owing to his persistence and bloody-mindedness (the latin ira translates to wrath, anger, fury), managed to obtain the large number of sediment cores required for my postdoctoral research.

Discussion

The present study provides the first record and description of a Lavareda female. Lavareda iramscotti sp. nov. females are characterized by two reflexed genital branches, as in the closely-related genus Rhadinema Cobb, 1920 and like the majority of other plectid taxa. The new species does not exhibit pronounced sexual dimorphism in features such as the buccal cavity, amphids, or tail. The diagnosis of the genus was modified to take into account the presence of body pores and epidermal glands in both sexes of the new species.

Lavareda coronatus was described from ca. 64 m water depth in the Colville Channel (Hauraki Gulf), North Island of New Zealand. Lavareda iramscotti sp. nov., also described from a New Zealand locality though from greater depth and further south, shows some close similarities with L. coronatus in the structure and size of the amphids and spicular apparatus, and both species have the same number of precloacal supplements. Some of the features of L. coronatus which differ from the new species, i.e., the apparent absence of body pores, cephalic setae, and precloacal seta in L. coronatus, may conceivably be the result of poor specimen condition (e.g., broken cephalic setae) or may have been missed in the original description. However, given the additional differences between Ditlevsen’s and my male specimens, notably in the ‘a’, ‘b’, and ‘c’ ratios, arrangement of the precloacal supplements, and number of testes, I consider it most likely that the two species are indeed different, even though they belong to a rare genus and were both described from New Zealand localities.

Conclusions

The present study provides the first record of a Leptolaimus species from the New Zealand region and the first description of a female specimen of the genus Lavareda. At least a dozen additional, yet to be described Leptolaimus species inhabit continental slope environments of New Zealand (D Leduc, 2012, unpublished data). Further research is needed to better characterise the diversity of this genus as well as that of other plectid nematodes across the region’s seabed habitats.

I thank two anonymous reviewers for providing constructive criticisms on the manuscript, Scott Nodder (NIWA) for obtaining the sediment samples, as well as the participants of NIWA voyages TAN0705 and TAN1103, and the officers and crew of RV Tangaroa.

Abbreviations

a body length/maximum body diameter

b body length/pharynx length

c body length/tail length

c′ tail length/anal or cloacal body diameter

cbd corresponding body diameter

L total body length

n number of specimens

V vulva distance from anterior end of body

%V V/total body length × 100

Additional Information and Declarations

Competing Interests

Author Contributions

Field Study Permissions

Data Availability

New Species Registration

The authors declare there are no competing interests.

Daniel Leduc conceived and designed the experiments, performed the experiments, analyzed the data, prepared figures and/or tables, authored or reviewed drafts of the paper, and approved the final draft.

The following information was supplied relating to field study approvals (i.e., approving body and any reference numbers):

The collection of sediment samples was conducted under Special permit 666 to NIWA granted by New Zealand’s Ministry for Primary Industries.

The following information was supplied regarding data availability:

The data used in this article is available in the Table and Figures.

The following information was supplied regarding the registration of a newly described species:

Publication LSID:

urn:lsid:zoobank.org:pub:4689626F-D42C-464A-BB0B-55D593DD3E71

Leptolaimus dififtinus sp. nov. LSID:

urn:lsid:zoobank.org:act:C4FE8E90-6481-4E1E-9780-561C4D1D8D9E

Lavareda iramscotti sp. nov. LSID:

urn:lsid:zoobank.org:act:79592D96-3AAE-4B67-9D23-14A158799216.

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
