# Peer review of "Two new nematode species (Plectida: Leptolaimidae, Rhadinematidae) from Chatham Rise, New Zealand"

_PeerJ, doi:10.7717/peerj.9923_

## Round 0.1 · original submission · Minor Revisions

Dear Dr. Leduc,

Thank you for submitting your paper to PeerJ. It now been reviewed and I have the following recommendations in line with the reviewers' reports.

I have received two reviews of your manuscript, and the reviewers conclude that the manuscript is interesting and will be a valuable contribution to our knowledge of the biodiversity of New Zealand fauna and biology of nematodes, but it needs a minor revision. The reviewers provided detailed comments, and I ask that you consider these carefully and correct the manuscript.

I look forward to receiving your revision.

Natalia Zhukova
Academic Editor, PeerJ

Reviewer 1 ·

Basic reporting

The manuscript is devoted to the description of two new to science species of nematodes from order Plectida. The findings are very interesting because representatives of this order rarely found in marine waters.
Manuscript is written in professional English and provided with sufficient background and context. Very good descriptions. Figures are of a very good quality.

Experimental design

no comment

Validity of the findings

no comment

Additional comments

But I still have some comments and reconsiderations.
Although line drawings are very good, there are not enough photographs. For Leptolaimus dififtinus please add photos of anterior end, .male tail, spicules and supplements at high magnifications.
For Lavareda iramscotti pleas add photos of anterior end with buccal cavity in focal plane, posterior end of male, copulative apparatus of male.

Line 113 Type locality
Can you also provide information on type of sediments, please?

Line 119 “body strongly curved ventrally”
Is it curved all the time or due to fixation with formalin?

Line 121 “alae ca 1.7 µm wide”
According to figure 1A the width of alae ca 2.5 µ, according to figure 3B the width of alae ca 3 µm. Please, clarify the width of alae.

Line 167 “can be differentiated from by the slightly shorter…”
Did you mean “can be differentiated from it by the slightly shorter…”?

Line 206 Type locality
Can you also provide information on type of sediments, please?

Line 217 “Inner and outer labial sensilla indistinct”
According to figure 4C and 4D outer labial sensilla papilliform.

Line 35 Allgen, 1932 absent in the references list.

Line 322 Allgen CA. 1930 Freilebende marine Nematoden…. Not cited in the text.

Line 882. Figure 1. There is no caption for figure 1B.

Line 390. Figure 4. There are no captions for figures 4E and 4G.

Reviewer 2 ·

Basic reporting

The article is very complete, well written, with correct and updated bibliography.
Very good drawings and photographs and very interesting.

Experimental design

The experimental design is used correctly for nematodes, so it is well used in the work

Validity of the findings

As I said before, it seems to me that the work is interesting and adds 2 new species to the studied area. It is very complete, well written and very well written.

Annotated reviews are not available for download in order to protect the identity of reviewers who chose to remain anonymous.

---

## Round 0.2 · accepted · Accept

Dear Dr. Leduc,
Thank you for submitting the revised version of your manuscript to PeerJ.
In the revised version the authors took into consideration all comments and remarks. I recommend to accept your manuscript for publication in PeerJ.